# Effects of Irrigation with Microcystin-Containing Water on Growth, Physiology, and Antioxidant Defense in Strawberry *Fragaria vulgaris* under Hydroponic Culture

**DOI:** 10.3390/toxins14030198

**Published:** 2022-03-07

**Authors:** Mohammed Haida, Fatima El Khalloufi, Richard Mugani, El Mahdi Redouane, Alexandre Campos, Vitor Vasconcelos, Brahim Oudra

**Affiliations:** 1Water, Biodiversity and Climate Change Laboratory, Faculty of Sciences Semlalia, Cadi Ayyad University, B.P. 2390, Marrakesh 40000, Morocco; mohammed.haida11@gmail.com (M.H.); richardmugani@gmail.com (R.M.); redouane.elmahdii@gmail.com (E.M.R.); oudra@uca.ac.ma (B.O.); 2Natural Resources Engineering and Environmental Impacts Team, Multidisciplinary Research and Innovation Laboratory, Polydisciplinary Faculty of Khouribga, Sultan Moulay Slimane University of Beni Mellal, B.P. 145, Khouribga 25000, Morocco; elkhalloufi.f@gmail.com; 3CIIMAR, Interdisciplinary Centre of Marine and Environmental Research, Terminal de Cruzeiros do Porto de Leixões, Av. General Norton de Matos, s/n, 4450-208 Porto, Portugal; acampos@ciimar.up.pt; 4Department of Biology, Faculty of Sciences, University of Porto, Rua do Campo Alegre, 4169-007 Porto, Portugal

**Keywords:** hydroponics, *Fragaria vulgaris* L., microcystins, irrigation water, plant growth, physiology, antioxidant defense

## Abstract

Over the last years, the use of artificial lakes and ponds to irrigate agricultural crops has been intensified and cultivation methods have been diversified. Hydroponics is a type of hydroculture which usually involves growing plants in an inert substrate, by using nutrient-enriched water to support plant growth. However, irrigating plants in hydroponic-based culture must be accompanied by monitoring the quality of irrigation water. The human health risks involved are mainly related to the proliferation of microcystin-producing cyanobacteria that contaminate water used for irrigation purposes. Strawberry (*Fragaria vulgaris* L.) is a widely cultivated plant of an increased economically importance worldwide. Its fruits provide essential elements for human nutrition; therefore, the study of its sensitivity to microcystins (MCs) is of paramount importance. The objective of this study was to evaluate the effects of MCs in irrigation water on the growth, physiology, and antioxidant defense system in *F. vulgaris.* In this study, strawberry seedlings at the three-leaf stage were grown in pots containing perlite under controlled conditions. Plants were exposed to a crude extract of *Microcystis aeruginosa* bloom at different concentrations of MCs (1, 5, 10, and 20 μg/L) for 60 days of exposure. The results showed that the highest concentrations of 10 and 20 μg/L induced a decrease in growth parameters. They resulted in root/shoot length decrease as well as number of leaves, roots/leaves dry and fresh weight. Furthermore, MCs reduced chlorophyll/carotenoid content, stomatal conductance, fluorescence, and total protein content of strawberry plants. At the same time, a significant increase in Malondialdehyde (MDA) (an indicator of lipid peroxidation), polyphenol, and sugar content were recorded in strawberry plants exposed to MCs at 5, 10, and 20 μg/L compared with the control. Additionally, superoxide dismutase (SOD), catalase (CAT), peroxidase (POD), Polyphenoloxydase (PPO), and ascorbate peroxidase (APX) activities significantly increased in plants under MCs exposure. The oxidative stress was higher in plants exposed to 10 and 20 μg/L of MCs from the second harvest (after 60 days of exposure) compared to those from the first harvest (after 30 days). Overall, the results obtained in this study indicate an increasingly negative effect of MCs on strawberry plants grown in hydroponics even at concentrations (10 and 20 µg/L). This effect is more damaging on the roots after exposure (60 days).

## 1. Introduction

Over the last decades, toxic cyanoblooms have been increasing in intensity and prevalence in freshwater bodies worldwide due to the joint effects of global warming and eutrophication [1,2,3,4,5]. It is estimated that over 75% of cyanobacterial blooms result in the production of a wide range of toxic secondary metabolites, called cyanotoxins, limiting the use of available water resources [6,7]. Among cyanobacterial toxins, microcystins (MCs) are the most widespread and worrisome class of cyanotoxins, which have attracted the attention of many researchers worldwide [8,9].

In Morocco, farmers resort to (hydroponic) cultivation in perlite to solve pH issues and provide plants’ (e.g., strawberries, blueberries, and raspberries) specific nutrient needs to improve crop yields. However, the problem is that irrigating farmland with MC-contaminated water sourced from reservoirs, ponds, and wells can pose a potential biohazard due to their accumulation in edible crops [10,11,12]. It’s known that MCs are water-soluble cyanopeptides, which facilitates their dispersal in irrigation water, resulting in their absorption by the root system and accumulation in different plant tissues. Their accumulation in the edible parts of agricultural crops results in food chain contamination causing adverse effects on animals and human health [10,11,12].

The main toxicity mechanism of MCs in plants and animals is well known, and consists of the specific inhibition of the catalytic subunit of proteins serine/threonine phosphatases 1 and 2A (PP1, PP2A) [13,14,15]. Moreover, MCs have various phytotoxic effects on: (a) seed germination [16], (b) plant growth and development [17,18], (c) photosynthetic performance [19,20,21], (d) and hormonal balance [22]. MCs are also responsible for metabolic-related impairments in plants, such as: (a) lipid peroxidation [23,24], (b) protein content decrease [20], and (c) oxidative stress [16].

After the exposure to MCs, plants generate and accumulate higher amounts of reactive oxygen species (ROS) which induces oxidative damage at the cellular level [10,25]. Plant antioxidant defense is mediated by several antioxidant enzymes that play a central role in regulating ROS-induced stress, such as catalase (CAT), peroxidase (POD), superoxide dismutase (SOD), polyphenol oxidase (PPO), and ascorbate peroxidase (APX) [17,18,25,26,27,28,29,30]. However, studies on oxidative stress of exposed edible crops to MC-contaminated water in hydroculture are still limited. In this type of culture, MCs are more bioavailable to plants, which may result in more phytotoxic impairments.

The objective of the present study was to examine the effects of long-term irrigation with an extract of the toxic *Microcystis aeruginosa* bloom on the growth, biochemical parameters, and antioxidant response of strawberry plants in hydroponic culture. In this regard, the results provide insight into the impact of microcystin-contaminated irrigation water on plants growing out of soils; and thus, contribute as a baseline for implementing legislation that regulates water quality in terms of microcystin concentrations in water used for plant irrigation.

## 2. Results

### 2.1. Effects of Microcystins on Strawberry Growth Parameters

The results presented in Table 1 show that both parts of the plant were sensitive to MCs. Regarding the root part, root length (RL), root dry weight (RDW), and root fresh weight (RFW) showed a decrease after exposure to MCs. Indeed, root dry weight (RDW) and root length (RL) decreased significantly (*p* < 0.05), by 46.42% and 25.84%, respectively, compared to the control after exposure to 20 μg/L of MCs for 60 days. Similarly, for aerial part parameters, a significant decrease (*p* < 0.05) in leaf length (LL) of 53% was recorded after exposure to 20 μg/L for 30 days, and of 68% for LDW after exposure to the same concentration for 60 days. These reductions are quite remarkable visually on plant roots and leaves. A two-way ANOVA showed a highly significant (*p* < 0.001) effect of concentration and exposure time (harvest) on growth parameters, except for root fresh weight (RFW) and leaf fresh weight (LFW) for the effect of exposure (Table 1). This indicates that the total effect of concentration on these two parameters exceeded the effect of exposure time. Similarly, the interaction between the total effects of the two factors on all measured parameters was non-significant, except for leaf length (LL) and leaf fresh weight (LDW), which were significantly influenced.

### 2.2. Effects of Microcystins on Strawberry Stomatal Conductivity and Chlorophyll Fluorescence

Exposure to cyanobacterial extracts caused a significant reduction compared with the control in stomatal conductivity. Decreases of 38.75%, 52.22%, and 76.06% after exposure to 5, 10, and 20 μg/L for 30 days, and of 36.49%, 51.79%, and 71.66%, respectively, after exposure to the same concentrations for 60 days were measured (Figure 1a). Fv/Fm values were slightly decreased with increasing toxin concentration, this decrease reached 18.53% and 22.53%, respectively, with concentrations of 10 and 20 μg/L, after 30 days of exposure; and reached 7.45% and 12.25%, respectively, after 60 days of exposure to the same concentrations (Figure 1b). Two-way ANOVA showed no significant interaction between the same MCs concentrations in the two harvests, except for the two concentrations of MCs (10 and 20 μg/L) in chlorophyll fluorescence (Figure 1).

### 2.3. Effects of Microcystins on Strawberry Pigments Contents

The exposure to cyanobacterial extracts containing MCs at various concentrations induced a decrease in chlorophyll a, b, total chlorophyll, and carotenoid contents in strawberry leaves during the exposure periods (Figure 2).

Chlorophyll content (a) showed a significant reduction with 10 and 20 μg/L of MCs compared with the control in both crops, decreasing by 57.72% after 30 days of exposure to 20 μg/L of MCs; and 40.89% after 60 days of exposure to the same concentration. Two-way ANOVA showed a significant interaction at (*p* < 0.001) between the same concentrations of MCs in both crops, except for the 10 μg/L concentration of MCs (Figure 2a). 

Chlorophyll (b) content showed highly significant (*p* < 0.05) decreases: by 35.52%, 40.02%, and 55% after 30 days of exposure to MCs at 5, 10, and 20 μg/L, respectively, and by 29.29%, 37.47%, and 43.98% after 60 days of exposure to MCs at 5, 10, and 20 μg/L, respectively, compared with the control. As the concentration of MCs increased, the chlorophyll (b) content decreased. No significant interactions were recorded between the concentrations in the two harvests (Figure 2b). 

The total chlorophyll content of strawberry leaves was reduced after 30 days of exposure to MCs, and further significantly reduced after 60 days of exposure to MCs, compared with the control. The interaction was highly significant between the two harvest periods (*p* < 0.001) in the 20 μg/L and 0 μg/L of MCs (Figure 2c).

Similarly, carotenoids decreased significantly with increasing concentration compared with the control. This decrease reached 44.31% and 64.65% after 30 days of exposure to MCs at 10 μg/L and 20 μg/L, respectively (Figure 2d). Similarly, for the second harvest, the carotenoid content decreased by 33.28% and 49.55% after 60 days of exposure to MCs at 10 μg/L and 20 μg/L, respectively. No significant interactions were recorded between the concentrations in the two harvests (Figure 2d).

### 2.4. Effects of Microcystins on Strawberry Lipid Peroxidation

According to Figure 3, an increase in MDA content was recorded in both plant organs during both exposure periods at all concentrations tested compared with the control group. Based on the results presented in Figure 3b, a significant increase (*p* < 0.05) in the percentage (86%) of lipid peroxidation was recorded in leaf cells after exposure to 20 μg/L of MCs for 60 days compared to the control. Whereas in roots, the percentage of lipid peroxidation reached its maximum (50%) compared with the control at the same concentration and harvest time (Figure 2a). According to the two-factor ANOVA, the interaction between the same concentrations of two harvests was highly significant, except for the concentration of 20 μg/L in roots and the control in leaves (Figure 3a,b).

### 2.5. Effects of Microcystins on Strawberry Phenolic Compounds 

The content of phenolic compounds in plant organs (leaves and roots) was high in the second harvest (Figure 4). The most remarkable of these results is that the content of phenolic compounds increased in roots by about 57%, 59%, and 62% after exposure for 60 days to 5, 10, and 20 μg/L of MCs, respectively, this increase did not show statistical significance between the 1, 5, and 10 μg/L of MCs (Figure 4a). Regarding leaves, significant increases of 59%, and 62% were observed after exposure to 10 and 20 μg/L of MCs for 60 days, compared to the control (Figure 4b). It is noted that the amount of phenolic compounds released in roots was higher than that in leaves. In addition, the interaction was highly significant between the same treatments used in both harvests in both organs (leaves and roots), except for the 20 μg/L concentration in roots.

### 2.6. Effects of Microcystins on Strawberry Soluble Sugar

Figure 5 shows the effect of MCs on soluble sugar content in strawberry roots and leaves. In roots, significant increases over the control in sugar content were recorded, by about 25%, 35%, 49%, and 50% after 30 days of exposure and 19.93%, 24.66%, 41.12%, and 51.44% after 60 days of exposure to 1, 5, 10, and 20 μg/L of MCs, respectively (Figure 5a). Soluble sugar content in leaves was significantly increased compared with the control by 57%, 58.37%, and 59% after 30 days of exposure to MCs, and by 43.60%, 47.70%, and 53.41% after 60 days of exposure to MCs at 1, 10, and 20 μg/L, respectively (Figure 5b). The interaction was not significant between the same concentrations at both harvests for root sugar content (Figure 5a). However, it was highly significant in leaves at 5, 10, and 20 µg/L MCs (Figure 5b).

### 2.7. Microcystins Effect on Total Protein and Antioxidant Enzyme Activities

In general, total protein content was reduced after irrigation of strawberry plants with water containing MCs (Figure 6). In contrast, an MC concentration of 20 μg/L in the irrigation water, decreased root protein content by up to 37% at the first harvest, and by 45% at the second harvest, compared with the control (Figure 6a). As with leaves (Figure 6b), protein content decreased significantly at 5, 10, and 20 μg/L of MCs in both harvests, relative to the control. The decrease in protein content was significant after chronic exposure of strawberry plants to MCs, regardless of the organ (root or leaf). Two-way ANOVA showed a significant interaction between the same MC concentrations and exposure duration (harvests) at 1 and 20 μg/L in roots (Figure 6a) and at 5 and 10 μg/L in leaves (Figure 6b).

CAT activity was significantly increased after 30 and 60 days of exposure to MCs at 5, 10, and 20 µg/L compared to the control. Figure 7. show that this increase was highly significant (85.44%) in the roots of the second harvest (60 days of exposure) at 20 µg/L MCs, whereas CAT activity in leaves increased by 79.75% after exposure to the same concentration for 60 days (Figure 7b). Two-way ANOVA of the interaction between the same concentrations of MCs in both crops was highly significant (*p* < 0.001) in all concentrations and in both crops and both organs (Figure 7a,b).

As shown in Figure 7c,d, at the second harvest (60 days-exposure to MCs), the changes in SOD activity in leaves and roots were also similar to those of CAT. The higher the concentration of MCs, the more the SOD activity was increased. The SOD activity of strawberry leaves increased significantly by 36.84% and 51.92% at 10 and 20 μg/L of MCs, respectively, after 60 days-exposure compared to the control. Whereas the SOD activity of strawberry roots increased more than that of leaves (31.69%, 36.84%, and 51.92%) for the same treatments (5, 10, and 20 μg/L of MCs). The interaction between the same concentrations and exposure times (harvests) was highly significant, especially at 5, 10, and 20 μg/L of MCs for both organs (Figure 7c,d). 

The data in Figure 7e,f show a negative effect of MCs on strawberry plants. This effect is reflected in an increase in POD enzyme activity. In contrast, POD activity at 5, 10, and 20 μg/L of MCs was significantly increased by 82%, 85%, and 90% in roots, and 92%, 94%, and 95% in leaves, respectively, compared with the control after 60 days of exposure. It appears that the interaction between the same concentrations of MCs in both harvests had no significant effect on the POD activity of strawberry plants in either roots or leaves (Figure 7e,f), except for the concentration of 1 μg/L in roots and 10 μg/L in leaves.

APX activity in strawberry roots and leaves increased significantly compared to the control under MCs exposure (Figure 7g,h). The highest concentrations of MCs caused changes in APX activity after 30 and 60 days of exposure. In roots, the percentage increase reached 47.99% after 30 days of exposure to MCs at 20 μg/L, and 52.52% after 60 days of exposure at the same concentration (Figure 7g). In leaves, the percentage increase reached 49% after 30 days of exposure to MCs at 20 µg/L, and 49.5% after 60 days of exposure at the same concentration (Figure 7h). The interaction between the same concentrations and harvest times was nonsignificant for most concentrations tested, except at 1 μg/L in roots (Figure 7g) and 10 μg/L in leaves (significant at *p* < 0.002) (Figure 7h).

PPO activity in strawberry roots increased significantly (*p* < 0.05) when plants were irrigated with MC-containing water at 1, 5, 10, and 20 μg/L compared to the control (Figure 7i). The leaves also showed a significant increase in PPO activity with the increase in MC concentrations. This increase (49.55%) was very remarkable at 20 μg/L after 60 days-exposure (Figure 7j). Highly significant interactions (*p* < 0.002) between the same concentrations in both harvests were recorded at 1 μg/L and 20 μg/L in roots, and at 10 μg/L and 20 μg/L in leaves. 

### 2.8. Principal Component Analysis

Principal component analysis (PCA) of growth parameters and pigment content (Figure 8a) showed that F1 and F2 components gave 96.37% variability, with 88.69% corresponding to F1 and 7.67% to F2. As for antioxidant enzymes and biochemical stress markers (Figure 8b), F1 and F2 components gave 93.82% variability, with 88.94% corresponding to F1 and 4.88% to F2. The PCA for *F. vulgaris* showed growth and pigment content (right) corresponding to the control (C) and low concentrations of MCs at 1 μg/L MCs (C1). Lower growth and pigment contents (left) correspond to high concentrations of MCs at 5 μg/L (C2), 10 μg/L (C3), and 20 μg/L (C4). On the vertical axis, C2 (5 μg/L) corresponds to intermediate growth and pigment content (Figure 8b). For antioxidant enzymes and biochemical stress markers, a detrimental effect (right) of the treatment with high concentrations, 10 μg/L (C3) and 20 μg/L (C4), of MCs was exhibited. Thus, the treatment with C2 corresponds to intermediate antioxidant enzyme activity and biochemical stress marker content (Figure 8b), whilst PCA shows no effect at 1 μg/L (C1) on antioxidant enzyme activity and stress markers in *F. vulgaris* plants.

## 3. Discussion

### 3.1. Strawberry Growth Parameters

MCs can reduce growth, induce oxidative stress and cause several physiological alterations in plants [20,31,32]. In this study, we carried out for the first time a study about the effect of MCs (*Microcystis aeruginosa*-bloom extract) on the growth, physiological, and biochemical parameters of strawberry plants grown in hydroponic culture. Strawberry plants were irrigated, via root spraying with MCs-containing nutrient solutions at 1, 5, 10, and 20 μg/L.

Our study revealed that the presence of MCs in irrigation water used in hydroponic culture mode increased the negative effect on various growth parameters of *F. vulgaris* plants, such as root and leaf length, root and leaf dry/fresh weight, and leaf number. This is in agreement with several studies that reported plant growth inhibition after MC exposure [17,19,24,25,26,28,29,33,34,35]. 

It was found that cyanobacterial extract containing concentrations of 10 and 20 μg/L of MCs cause a reduction in the growth of a plant of economic importance (*F. vulgaris*). This reduction results in a decrease in root and leaf length. El Khalloufi et al. (2011) [16] and Pereira et al. (2009) [35] recorded an inhibition of root growth of *Medicago sativa* and *Lactuca sativa* after exposure to MCs. In addition, Chen et al. (2012) [30] reported that high concentrations (2 mg/L of MCs) inhibited root elongation, crown root formation, and lateral root development for *Brassica rapa* (Chen et al., 2012) [28], a decrease in root and leaf length after exposure for 6 days to 1 and 10 μg/L of MCs from either cyanobacteria extract or the pure toxin for *Lepidium sativum* (Gehringer et al., 2003) [36]. However, the results of this study showed a slight decrease in the fresh and dry weight of leaves and roots and the number of plant leaves of *F. vulgaris* when increasing the concentration of MCs in the used irrigation water, similarly, a high concentration (100 μg/L) of MCs decreased the fresh weight of the plant *Lactuca sativa* (Freitas et al., 2015) [31], decreased fresh weight of *Lepidium sativum* seedlings [36], and decreased of root dry weight after exposure of *Vicia faba* to 100 μg/L [29].

According to the results obtained on the effect of MCs on the growth parameters of *F. vulgaris*, this mode of culture (hydroponics) increases the contact between MCs and plants, which increases the effect of MCs on the plant of *F. vulgaris*, even at low concentrations. This effect is expressed by the inhibition of protein phosphatases, because microcystins render inactive the irreversible binding of protein phosphatases, in particular PP1 and PP2A [13,37], these protein phosphatases are well known to be involved in the regulation of several physiological processes by dephosphorylation of regulatory proteins. Inhibition of these proteins in the plant results in leaf malformations, histological changes, and a delay in root organ differentiation and vascular cylinder formation with inhibition of lateral primordial root formation [19]. In general, inhibition of root and leaf growth is a useful indicator of toxicity because it can directly affect plant survival and growth, and thus on strawberry fruit yield.

### 3.2. Stomatal Conductivity and Chlorophyll Fluorescence

The results obtained in this study indicate for the first time a decrease in stomatal conductance of *F. vulgaris* leaves after exposure to MCs. Indeed, these results are in contradiction with a study by Bittencourt-Oliveira et al. (2016) [21] that reported an increase in stomatal conductance after irrigation of *Lactuca sativa* with water contaminated with different concentrations of MCs. Indeed, MCs can reduce stomatal conductance, which decreases CO_2_ availability and assimilation [38]. Regarding leaf fluorescence (Fv/Fm ratio), a slight decrease was shown mainly at the high concentration (20 μg/L). Similarly, exposure of *Lycopersicon esculentum* to 22.24 µg/mL of MCs for 30 days caused negative effects on photosystem II activity, reflected by a decrease in Fv/Fm fluorescence [17].

### 3.3. Pigment Contents

The study of variations in pigment contents (chlorophylls and carotenoids) is an indicator of plant health. These pigments play very important roles in photosynthesis which directly influences growth, and the plants’ development [39,40]. Any change in their content can indicate an imbalance and disturbances in the photosynthesis process [41]. This study confirms the negative effect of cyanobacterial MCs on the photosynthetic activity of *F. vulgaris* after chronic exposure (60 days), this effect is reflected in the reduction of chlorophyll a, b, total, and carotenoids. In several studies carried out on this subject, decreases in chlorophyll (a and b) after irrigation with an extract of MCs in the leaves of *Zea mays* and *Lens esculenta* and *Triticum aestivum* were reported [42,43]. Moreover, a decrease in carotenoid content after exposure of strawberry plants to MCs was recorded, which is in disagreement with the results obtained by Pereira et al. (2017) [44] which recorded no variation of this parameter in plants *Coriandrum sativum* L. and *Petroselinum crispum* L. In general, this study suggests that the synthesis of chlorophyll and carotenoids in *F. vulgaris* was negatively affected by MCs in irrigation water, resulting in slower plant growth. This may be due to malfunction of PSII, inhibition of electron transport-related to reaction center excitation, or may be attributed to low mineral uptake [45].

### 3.4. Effects of Microcystins on Strawberry Lipid Peroxidation

MDA is used as an indicator of lipid peroxidation under oxidative stress inducers. In the present study, a marked increase in MDA levels in strawberry leaves and roots after chronic exposure to MCs. This result is similar to the study of Pflugmacher et al. (2006) [24] who reported an increase in MDA content in *Medicago sativa* plant cells after exposure to 0.5 μg/L of MCs. This increase indicates the inability of the antioxidant defense system to protect plants from excessive ROS production and intracellular damage [46].

### 3.5. Strawberry Phenolic Compounds

Phenolic compounds play a major role in plant adaptation to abiotic and biotic stresses [47]. An increase in phenolic compound content was reported after chronic exposure to MCs especially in the root part of the strawberry. Similar results have been reported for *Medicago sativa* exposed to 11.12 and 22.24 µg/mL and *Vicia faba* to 50 and 100 μg/L of MCs [16,29], as well as a high accumulation of phenolic compounds *Raphanus sativus* and *Daucus carota,* was recorded after irrigation with MC-contaminated water [48]. Generally, an increase in this secondary metabolite is a defense strategy under stress, it acts in the detoxification of free radicals in plants treated with cyanotoxins to maintain the vital functions of plants [25,49].

### 3.6. Strawberry Soluble Sugar

Soluble sugars in strawberry leaves and roots increased with increasing concentrations of MCs and with increasing exposure time (60 days), especially in leaves, where the increase was very significant. No studies on the effect of MCs on this parameter have been performed, but there is most probably a relationship between the increase in soluble sugars in the plant cells and the deterioration of cell membranes. This increase may be a reaction of the plant as an osmotic adjustment to oxidative stress [50].

### 3.7. Total Proteins and Antioxidant Enzymes Activities

In the present study, a decrease in total protein content was recorded in strawberry leaves and roots after chronic exposure to MCs. The protein content of plants stressed by exposure to MCs has remained incomprehensible until now. A decrease in total protein content in *Lactuca sativa* leaves at high concentration has been shown [21], and a significant increase in protein content in *Medicago sativa* leaves and roots after exposure to MCs has been recorded [16]. Our results can be explained by the ability of MCs to inhibit protein synthesis in plants and give priority to the biosynthesis of important antioxidant enzymes to help protect the plant against the effect of ROS [51,52].

Excessive ROS release is an indicator of oxidative stress inducing several changes in cell membrane structure and function by attacking unsaturated fatty acids of cell membrane lipids, triggering the lipid peroxidation chain reaction, and leading to a decrease in plant biomass accumulation [53,54]. The intervention of the antioxidant defense system is an important regulatory mechanism in plants for abiotic stress resistance and control of ROS homeostasis [27,55,56]. Several studies have reported the effects of MCs on plants grown in soil. This study showed for the first time an increase in the enzymatic activity of CAT, SOD, POD, APX, and PPO in hydroponically grown plants. This increase is very important at the level of the roots which are in direct contact with the MCs compared to the leaves. Similarly, an increase in CAT, SOD, and POD activity in seedling cells of *Cucumis sativus* and *Oryza sativa* after exposure to 100 μg/L for 14 days has been observed [32], an increase in POD activity after exposure of *Eruca sativa* to toxic extracts of MCs for 15 days [57], an increase in SOD, CAT and APX activity of *Lactuca sativa* and *Brassica rapa* exposed to high concentrations of MCs (30, 400, 6000, and 6400 μg/L) [27,28]. In the same context, the activity of POD, PPO, and CAT in leaves and roots of *Medicago sativa* and *Vicia faba* increased after exposure to 10, 20, 50, and 100 μg/L of MCs [29,58]. To solve the problems of pH and meet the specific nutrient needs of plants (strawberry, blueberry, and raspberry...) to obtain a good yield, farmers resort to this type of culture (hydroponics). However, the problem that arises is that these plants are irrigated by water from dams contaminated by MCs. The negative effect of these MCs in this mode of cultivation can be explained on one hand by the chronic exposure to MCs, and on the other hand by the absence of soil particles that can adsorb the MCs and the absence of soil microorganisms that may be involved in different MC degradation pathways that play an important role in increasing plant contact with microcystins. From all these elements, we can say that the degree of influence of MCs is related to the exposure time, the concentration of MCs, the plant organ studied, and type of substrate used in cultivation.

## 4. Conclusions

Overall, we conclude that MC-induced stress caused growth and physiological impairments in strawberry plants grown in hydroponic culture. Furthermore, the results showed that irrigation with MC-contaminated water resulted in growth reduction at the highest concentrations, and a decrease in the photosynthesis parameters. We also noted a decrease in total protein content and an increase in soluble sugars as well as lipid peroxidation. MC-induced stress was accompanied by a high release of polyphenols and a significant increase in CAT, SOD, POD, PPO, and APX activities in both leaves and roots at 1, 5, 10, and 20 μg/L of MCs. To our knowledge, this is one of the first reports on the toxicity of MCs on a plant of high nutritional and commercial value such as strawberry in hydroponics. These results provide insight into the biohazard of MC-polluted irrigation water on edible crops and public health. Furthermore, these findings imply the need to develop new methods and strategies for the removal of MCs and their producers from water bodies used for irrigation purposes.

## 5. Materials and Methods

### 5.1. Microcystins Extraction and Quantification

The *Microcystis aeruginosa* bloom used for the crude extract containing Microcystins preparation was collected in October 2010 from the Lalla Takerkoust reservoir located at Marrakesh, Morocco (31° 36′ N, 8° 2′ W, 664 m). Briefly, for the extract preparation, 250 mg of freeze-dried cyanobacterial biomass was homogenized in liquid nitrogen, sonicated in an ice bath for 5 min (42 kHz) to release intracellular MCs, and then centrifuged at 10,000× *g* for 15 min. After that, the extraction was repeated twice. The supernatants were collected and stored at −20 °C until use. The bloom characterization was performed as described by El Khalloufi et al. (2013) [58]. The total Microcystin concentration was previously determined and quantified using the protein phosphatase type 2A inhibition assay as described by Bouaïcha et al. (2001) [59]. The test is based on the dephosphorylation of para-nitrophenylphosphate (*p*-NPP) and its transformation into a colored product: para-nitrophenol. The enzyme activity is then determined based on the colored compound production at 405 nm. Thereby, the inhibitory effect of Microcystins on the dephosphorylation of *p*-NPP by PP2A enzyme was measured. The total amount of Microcystins in the collected bloom was then expressed as mg MC-LR equivalent g^−1^ DW. The PP2A analysis of the M. aeruginosa flower indicated a total MC concentration of 11.5 mg MC-LR g^−1^ DW equivalent. In addition, qualitative analysis performed using a high-performance liquid chromatography (HPLC) system for the identification of MCs variants revealed the predominance of the MC-LR variant (at 98%) [58]. For the conducted experiment, concentrations of MCs at 1, 5, 10, and 20 μg/L were then prepared (Figure 9).

### 5.2. Experimental Setup

Three-leaf strawberry (*F. vulgaris*) seedlings were acclimated for 7 days and then transplanted into 1.5 L plastic pots containing perlite (a substrate widely used in hydroponic culture). Plants were grown in a controlled growth chamber at 20 ± 1 °C with a light intensity of 200 μmol. m^−2^. s^−1^, a photoperiod of 13/11 h (day/night), and humidity of 70–80%. The experiment was set up in a completely randomized design with 6 replicates per treatment (one plant per pot). MCs extract was diluted with distilled water or a modified Hoagland nutrient solution [60] (Table 2) to final concentrations: C = 0 μg/L, C1 = 1 μg/L, C2 = 5 μg/L, C3 = 10 μg/L, and C4 = 20 μg/L. Plants were watered with nutrient solution and water containing MCs at 2-day intervals, alternately. Plant biomass was harvested at two periods (first month: Harvest 1 and second month: Harvest 2) for the determination of the parameters cited below (Figure 10).

### 5.3. Plant Harvest and Growth Parameters

Plants corresponding to Harvest 1 and 2 were divided into leaves and roots. Morphometric parameters were determined including number of leaves (LN), leaves and root length (LL and RL respectively), root/leaves (R/L) fresh and dry weights (FW and DW respectively). 

### 5.4. Physiological Parameters

#### 5.4.1. Stomatal Conductance and Chlorophyll Fluorescence

Stomatal conductance (gs) was measured at noon in full light on two leaves per plant with four replicates for each treatment using a portable porometer (Leaf Porometer LP1989, Decagon Device, Inc., Washington, DC, USA). Chlorophyll fluorescence was recorded on dark-adapted leaves (30 min) using a portable fluorometer (Opti-sciences OSI 30p, Hudson, NY, USA). Chlorophyll fluorescence was estimated by the ratio Fm-F0/Fm where F0 is the zero-level fluorescence and Fm is the maximum fluorescence [61].

#### 5.4.2. Pigments Determination

Chlorophylls and carotenoids were extracted from leaf samples in acetone as described by Upadhyaya et al. (2019) [62]. Approximately 0.5 g of leaves were extracted with 5 mL of 95.5% acetone. The concentrations of chlorophyll a (*Chl a*), chlorophyll b (*Chl b*), and carotenoids (Caro) were measured at 662, 644, and 470 nm, respectively, using a UV-visible spectrophotometer (Cary 50 Scan, Australia). The concentrations of *Chl a*, *Chl b*, total chlorophyll, and carotenoids in leaf tissue were calculated according to the following equations and expressed as mg. g^−1^ of fresh mass:$$\mathrm{Chl}\;a=9.784\;DO_{662}-0.99\;DO_{644}$$
$$\mathrm{Chl}\;b=21.42\;DO_{644}-4.65\;DO_{662}$$
Total chlorophyll=Chl a+Chl b
Caro=1000 DO470−1.90 Chla−63.14 Chl b214

#### 5.4.3. Total Sugar Content

To determine soluble sugar content, 0.1 g of leaves or roots were homogenized with 4 mL of ethanol (80% *v/v*), and the homogenate was centrifuged at 5000× *g* for 10 min. 1 mL of supernatant was mixed with 1 mL of phenol solution (5%) and 5 mL of concentrated sulfuric acid. After 5 min, the optical density was measured at 485 nm in a UV-visible spectrophotometer (Square 50 Scan, Australia). Soluble sugar content was determined using glucose solution as a standard [63].

### 5.5. Plant Antioxidant Defense

#### 5.5.1. Malondialdehyde Determination

MDA is a product of lipid peroxidation, was determined by the thiobarbituric acid (TBA) reaction as described by Savicka et al. (2010) [64]. Briefly, 0.2 g of leaves or roots were homogenized with 2 mL of 0.1% trichloroacetic acid (TCA) and centrifuged at 14,000× *g* for 15 min. After centrifugation, 1 mL of the supernatant was mixed with 2.5 mL of 0.5% TBA in 20% TCA. The mixture was heated to 95 °C for 30 min, and then rapidly cooled in an ice bath. Then it was centrifuged at 10,000× *g* for 30 min. The absorbance of the supernatant at 532 nm was read and the value of the nonspecific absorbance at 600 nm was subtracted using an absorbance extinction coefficient (155 mM^−1^.cm^−1^).

#### 5.5.2. Total Phenolic Compounds Determination

Phenolic compounds are secondary metabolites and have a high antioxidant capacity. Determination of phenolic compound content was performed according to the method described by Kähkönen et al. (1999) [65] with slight modifications. Approximately 0.2 g of *F. vulgaris* roots or leaves per treatment were homogenized in 95% methanol. The homogenate was centrifuged at 30,000× *g* for 10 min. 1 mL aliquots of the extracts were mixed with 1 mL of 1 N Folin-Ciocalteau reagent and 1 mL of 10% sodium carbonate, and the mixture was incubated for 1 h at 35 °C. The absorbance was measured at 530 nm and total phenol content was expressed as gallic acid equivalents in milligrams per gram of dry matter.

#### 5.5.3. Antioxidant Enzymes Activity 

For enzyme extracts and assays, 0.5 g of fresh leaves or roots were ground, in a cold mortar, with liquid nitrogen and homogenized in 5 mL of a solution containing 50 mM potassium phosphate buffer (pH 7.0), 5% (*w/v*) polyvinylpolypyrrolidone, and 0.1 mM ethylenediaminetetraacetic acid (EDTA). Afterward, the extracts were centrifuged at 4 °C for 20 min at 12,500× *g*, and the supernatants were used to measure total protein and enzyme activities [66].

Protein determination was performed by the Bradford method (1976) [67]. Volumes of 2 mL of diluted enzyme extract were taken in tubes and added to 2 mL of Bradford’s reagent. After 4 min of incubation, the optical density was measured at 595 nm. Protein content was determined using a standard curve established by the bovine serum albumin (BSA) standard solutions and expressed as mg.g^−1^ DW.

SOD activity was determined by measuring the inhibition of photochemical reduction of tetrazolium nitrobleu according to the method described by Beyer and Fridovich (1987) [68], with some modifications. A 3 mL reaction mixture contained 100 mM phosphate buffer (pH 7.8), 55 mM methionine, 0.75 mM tetrazolium nitro blue, 0.1 mM riboflavin, and 50 μL of the enzyme extract. The reaction was initiated by placing the tubes under 20 W fluorescent lamps at 25 °C after Riboflavin addition. The absorbance was read after 15 min at 560 nm.

For CAT activity, the reaction mixture consisted of 2.6 mL of 50 mM phosphate buffer (pH 7.0), 200 µL of 15 mM hydrogen peroxide, and 200 µL of the enzyme extract. The decomposition of H_2_O_2_ was monitored at 240 nm (ε = 39.4 mM cm^−1^). CAT activity was expressed as μmol of H_2_O_2_ reduced. mg protein^−1^ min^−1^ [69].

POD activity was determined according to the method of Fidalgo et al. (2013) [70]. The reaction mixture contained 50 mM PBS, 0.1% guaiacol, 2% H_2_O_2,_ and 1 mL of enzyme solution. Guaiacol oxidation was measured at 470 nm (ε = 26.6 mM cm^−1^) for 3 min. POD activity was expressed per µmol guaiacol oxidized min^−1^.mg^−1^ protein.

APX activity was evaluated according to the method of Nakano and Asada, (1981) [71], with some modifications. The reaction mixture contained 1 mL of ice-cold 50 mM potassium phosphate buffer (pH 7.8), 2 mM ascorbic acid and 5 mM EDTA, 0.1 mM H_2_O_2_, and 100 µL of the enzyme extract. The decomposition of ascorbic acid was monitored at 290 nm. APX activity was expressed per µmol of oxidized ascorbic acid min^−1^ mg^−1^ protein.

PPO activity was determined according to the method of Wojdyło et al. (2013) [72] by following the oxidation of catechol at 420 nm for 3 min. The activity was measured in 3 mL of a reaction mixture consisting of 2.7 mL of 0.1 M catechol dissolved in 100 mM sodium phosphate buffer (pH 5.5) and 0.3 mL of enzyme extract. The activity of PPO was expressed as mg^−1^ protein. min^−1^.

### 5.6. Data Analysis

Analysis of growth and physiological parameters were performed in sex replicates per treatment. Results are given as mean ± standard error (SE). Differences between treatments were assessed by one-way ANOVA and means were compared by Tukey’s HSD test. Significant differences at *p* < 0.05 are indicated by different letters. A two-factor analysis of variance (ANOVA) was performed to assess the significant effects of treatments and exposure time and their interaction. For that, data were log transformed to fulfill ANOVA assumptions of normality. Significant differences in two-way ANOVA were performed at, *p* < 0.05, ** *p* < 0.01, and *** *p* < 0.001 for different factors. ANOVA was performed using SPSS version 22.0 statistical software. Growth parameters, biochemical stress markers, and antioxidant enzymes and their correlation with treatments were subjected to principal analyses (PCA) using XLStat software.

## Figures and Tables

**Figure 1 toxins-14-00198-f001:**
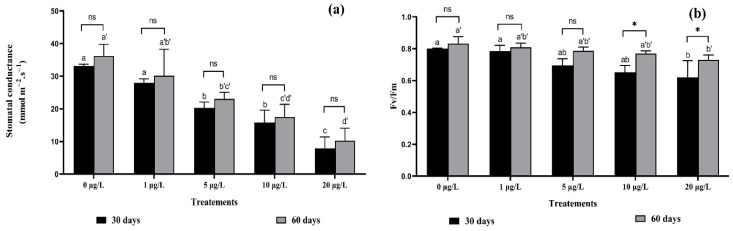
(**a**) Stomatal conductance, (**b**) Ratio Fv/Fm, in *F. vulgaris* leaves exposed to MCs for 30- and 60-days exposure. The bars represent the mean values. The error bars represent the standard deviations. One-way ANOVA indicates significant difference at (*p* < 0.05) between the control and the treatments for each harvest (30 days or 60 days exposure time) by different letters (letter “x” for 30 days and “x’” for 60 days). The two-way ANOVA indicates ns, not significant, * *p* < 0.05 significant difference between exposure time for each treatment group, ns: no significant difference.

**Figure 2 toxins-14-00198-f002:**
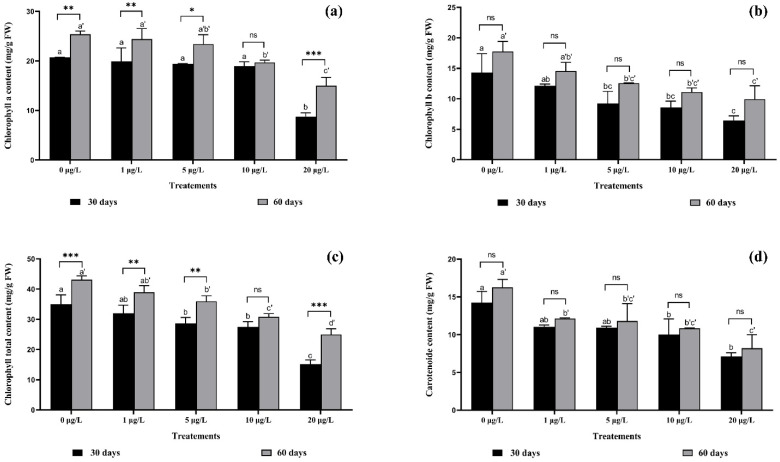
(**a**) Chlorophyll a, (**b**) chlorophyll b, (**c**) total chlorophyll, and (**d**) carotenoids content in *F. vulgaris* leaves exposed to MCs for 30- and 60-days exposure. The bars represent the mean values. The error bars represent the standard deviations. One-way ANOVA indicates significant difference at (*p* < 0.05) between the control and the treatments for each harvest (30 days or 60 days exposure time) by different letters (letter “x” for 30 days and “x’” for 60 days). The two-way ANOVA indicates * *p* < 0.05, ** *p* < 0.01, or *** *p*< 0.001 significant difference between exposure time for each treatment group. ns: no significant difference.

**Figure 3 toxins-14-00198-f003:**
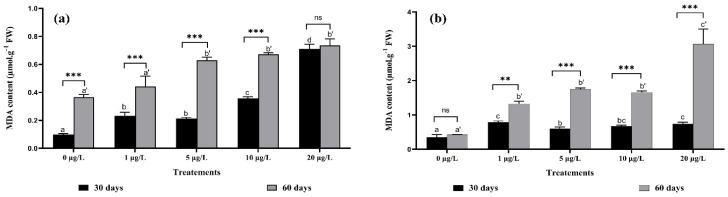
MDA content in: (**a**) roots, (**b**) leaves of *F. vulgaris* exposed to MCs for 30 and 60 days. The bars represent the mean values. The error bars represent the standard deviations. One-way ANOVA indicates significant difference at (*p* < 0.05) between the control and the treatments for each harvest (30 days or 60 days exposure time) by different letters (letter “x” for 30 days and “x’” for 60 days). The two-way ANOVA indicates ** *p* < 0.01, or *** *p* < 0.001 significant difference between exposure time for each treatment group. ns: no significant difference.

**Figure 4 toxins-14-00198-f004:**
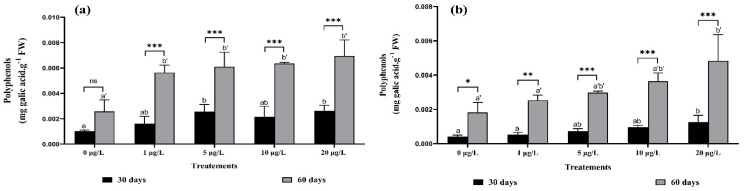
Polyphenols content in: (**a**) roots, (**b**) leaves of *F. vulgaris* exposed to MCs for 30 and 60 days. Bars represent mean values. Error bars represent standard deviations. One-way ANOVA indicates significant difference at (*p* < 0.05) between the control and the treatments for each harvest (30 days or 60 days exposure time) by different letters (letter “x” for 30 days and “x’” for 60 days). The two-way ANOVA indicates * *p* < 0.05, ** *p* < 0.01, or *** *p* < 0.001 significant difference between exposure time for each treatment group. ns: no significant difference.

**Figure 5 toxins-14-00198-f005:**
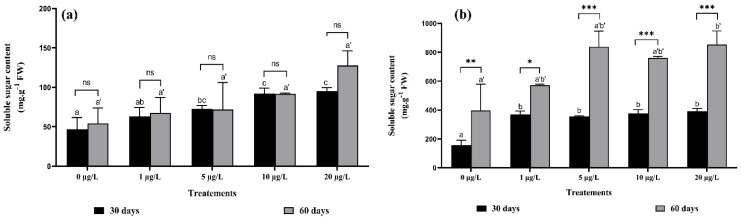
Soluble sugar content in: (**a**) roots, (**b**) leaves of *F. vulgaris* exposed to MCs for 30 and 60 days. Bars represent mean values. Error bars represent standard deviations. One-way ANOVA indicates significant difference at (*p* < 0.05) between the control and the treatments for each harvest (30 days or 60 days exposure time) by different letters (letter “x” for 30 days and “x’” for 60 days). The two-way ANOVA indicates * *p* < 0.05, ** *p* < 0.01, or *** *p* < 0.001 significant difference between exposure time for each treatment group. ns: no significant difference.

**Figure 6 toxins-14-00198-f006:**
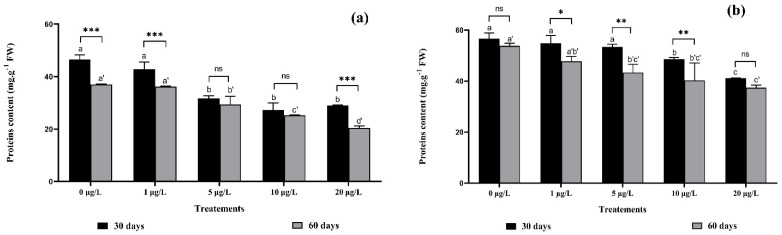
Total protein content in: (**a**) roots and (**b**) leaves of *F. vulgaris* exposed to MCs for 30 and 60 days. Bars represent mean values. Error bars represent standard deviations. One-way ANOVA indicates significant difference at (*p* < 0.05) between the control and the treatments for each harvest (30 days or 60 days exposure time) by different letters (letter “x” for 30 days and “x’” for 60 days). The two-way ANOVA indicate * *p* < 0.05, ** *p* < 0.01 or *** *p* < 0.001 significant difference between exposure time for each treatment group. ns: no significant difference.

**Figure 7 toxins-14-00198-f007:**
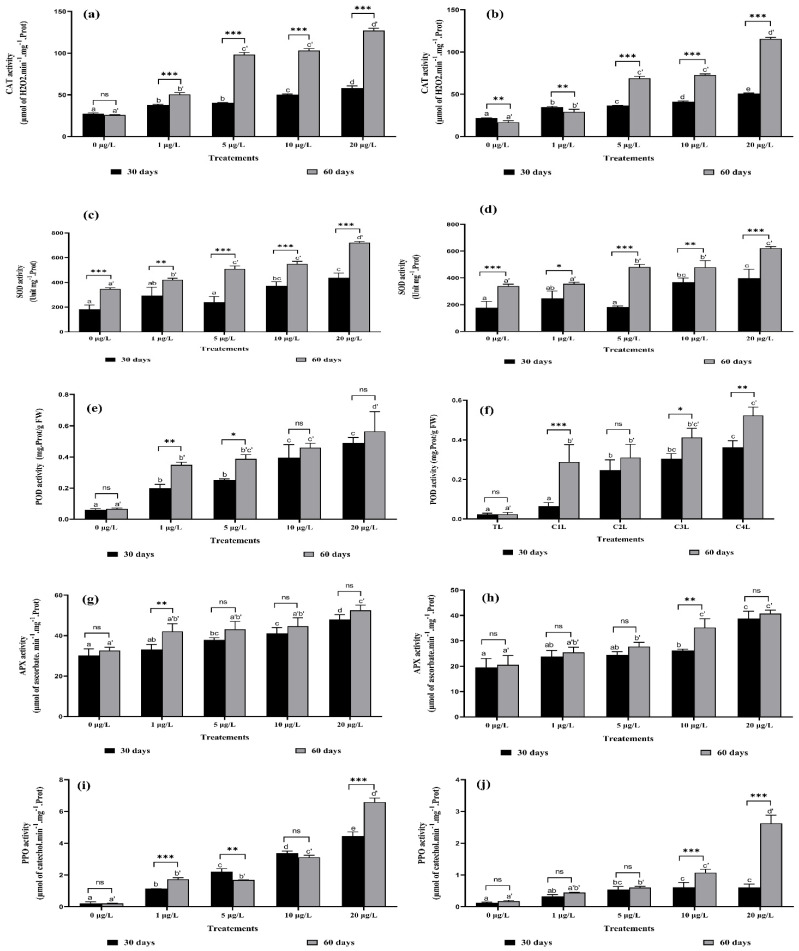
CAT activity in: (**a**) roots and (**b**) leaves; SOD activity in: (**c**) roots and (**d**) leaves; POD activity in: (**e**) roots and (**f**) leaves; APX activity in: (**g**) roots and (**h**) leaves; and PPO activity in (**i**) roots and (**j**) leaves of *F. vulgaris* exposed to MCs for 30 and 60 days. Bars represent mean values. Error bars represent standard deviations. One-way ANOVA indicate significant difference at (*p* < 0.05) between the control and the treatments for each harvest (30 days or 60 days exposure time) by different letters (letter “x” for 30 days and “x’” for 60 days). The two-way ANOVA indicates * *p* < 0.05, ** *p* < 0.01, or *** *p* < 0.001 significant difference between exposure time for each treatment group. ns: no significant difference.

**Figure 8 toxins-14-00198-f008:**
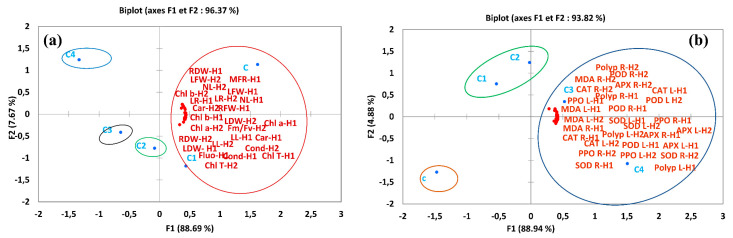
Principal component analysis of the studied parameters according to the treatments applied to strawberry (*F. vulgaris),* (**a**): growth parameters and pigment content, (**b**): antioxidant enzymes and biochemical parameters, C: control, C1: 1 μg/L of MCs, C2: 5 μg/L of MCs, C3: 10 μg/L of MCs, C3: 20 μg/L of MCs, RL: root length, DWR: root dry weight, DWL: leaves dry weight, LL: leaves length, LFW: leaves fresh weight, LDW: leaves dry weight, Chl a: chlorophyll a, Chl b: chlorophyll b, Chl T: chlorophyll total, Carot: carotenoid, Condu: stomatal conductance, Fm/Fv: chlorophyll fluorescence, H1: harvest 1 (30 days of exposure to MCs), H2: harvest 2 (60 days of exposure to MCs), MDA R: lipid peroxidation in roots, MDA L: lipid peroxidation in leaves, polyp R: polyphenols in roots, polyp L: polyphenols in leaves, SOD R: superoxide dismutase in roots, SOD L: superoxide dismutase in leaves, CAT L: catalase in leaves, APX R: ascorbate peroxidase in roots, APX L: ascorbate peroxidase in leaves, PPO R: polyphenoloxidase in roots, PPO L: polyphenoloxidase in leaves.

**Figure 9 toxins-14-00198-f009:**
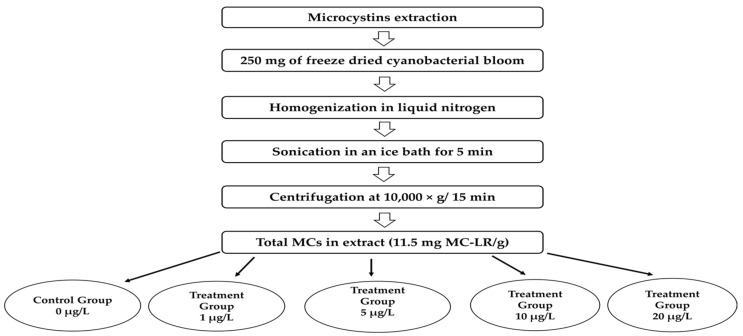
Experimental design for extraction of MCs from the cyanobacterial bloom of *Microcystis aeruginosa* used in the five independent exposure experiments: 0, 1, 5, 10, and 20 µg/L MCs. Six pots per treatment/control group were used.

**Figure 10 toxins-14-00198-f010:**
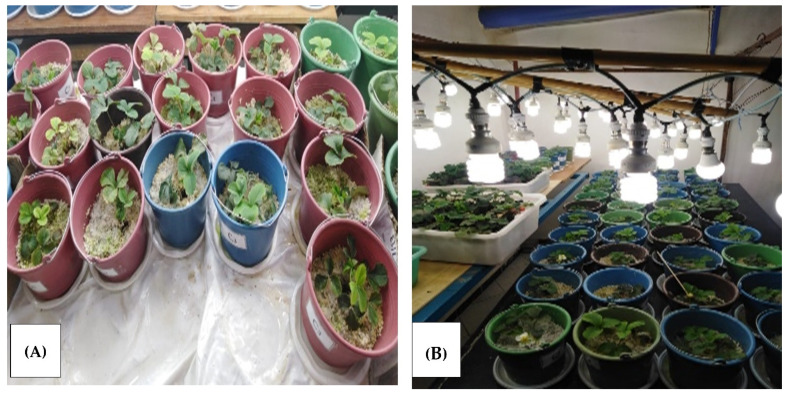
Strawberry experiment under controlled environmental conditions after (**A**) 30 and (**B**) 60 days of experiment.

**Table 1 toxins-14-00198-t001:** Effect of microcystins on plant growth parameters.

Treatments		RL (cm)	RDW (g)	RFW (g)	NL	LL (cm)	LDW (g)	LFW (g)
Harvest 1	Control	24.00 ± 0.50 ^a^	6.56 ± 0.34 ^a^	13.33 ± 1.81 ^a^	6.83 ± 0.67 ^a^	20.57 ± 0.36 ^a^	4.38 ± 0.13 ^a^	14.27 ± 0.36 ^a^
	1 µg/L	19.83 ± 0.59 ^a^	3.40 ^b^ ± 0.24 ^b^	11.33 ± 1.49 ^a^	6.33 ± 0.61 ^ab^	16.61 ± 0.74 ^b^	3.64 ± 0.07 ^b^	12.82 ± 0.40 ^b^
	5 μg/L	18.83 ± 0.29 ^a^	3.03 ± 0.60 ^b^	11.08 ± 1.04 ^a^	5.33 ± 0.52 ^b^	14.53 ± 0.31 ^c^	2.60 ± 0.03 ^c^	12.67 ± 0.60 ^b^
	10 μg/L	17.17 ± 0.07 ^a^	2.93 ± 0.59 ^b^	11.17 ± 0.85 ^a^	4.67 ± 0.43 ^b^	11.64 ± 0.24 ^d^	2.63 ± 0.17 ^c^	12.57 ± 0.32 ^b^
	20 μg/L	18.50 ± 0.15 ^a^	3.07 ± 0.72 ^b^	9.67 ± 0.94 ^a^	4.60 ± 0.63 ^b^	9.65 ± 0.26 ^e^	1.51 ± 0.06 ^d^	12.63 ± 0.14 ^b^
Harvest 2	Control	29.67 ± 0.33 ^a’^	6.80 ± 1.08 ^a’^	15.33 ± 2.91 ^a’^	8.83 ± 0.70 ^a’^	25.65 ± 1.02 ^a’^	5.71 ± 0.12 ^a’^	15.50 ± 0.61 ^a’^
	1 μg/L	25.00 ± 0.44 ^a’b’^	5.65 ± 0.71 ^a’b’^	13.67 ± 1.58 ^a’^	6.00 ± 0.99 ^a’^	23.42 ± 0.45 ^a’b’^	4.62 ± 0.01 ^b’^	13.10 ± 0.85 ^a’^
	5 μg/L	24.17 ± 1.08 ^a’b’^	5.50 ^a’b’^ ± 0.65 ^a’b’^	11.67 ± 2.39 ^a’^	7.00 ± 0.52 ^a’^	20.80 ± 0.65 ^b’^	3.28 ± 0.20 ^c’^	12.94 ± 1.15 ^a’^
	10 μg/L	22.17 ± 0.91 ^b’^	4.20 ± 0.15 ^b’^	11.83 ± 0.77 ^a’^	6.33 ± 1.07 ^a’^	15.77 ± 0.28 ^c’^	2.47 ± 0.15 ^c’d’^	12.58 ± 0.86 ^a’^
	20 μg/L	22.00 ± 0.95 ^b’^	3.64 ± 0.23 ^b’^	10.50 ± 1.08 ^a’^	6.00 ± 0.63 ^a’^	12.91 ± 0.82 ^c’^	1.80 ± 0.38 ^d’^	12.37 ± 0.77 ^a’^
**Significance**								
The total effect of concentration (1)		***	**	**	**	***	***	***
The total effect of harvest (2)		***	**	ns	***	***	***	ns
(1) × (2)		ns	ns	ns	ns	**	**	ns

The values are denoted as Mean ± Standard Error (*n* = 6). Values with the same letters within each column indicate no significant difference (*p* < 0.05) by Tukey test for each harvest: ns, not significant; ** *p* < 0.01, moderate significant differences; *** *p* < 0.001, highly significant differences between exposure time for each treatment group using two-way ANOVA and Tukey test. RL: root length, RDW: root dry weight, RFW: root fresh weight, NL: number of leaves, LL: leaves length, LDW: leaves dry weight, SFW: shoot fresh weight.

**Table 2 toxins-14-00198-t002:** Composition of the nutrient solution of Hoagland [60].

Macroelements	Quantity (in g/L)	Microelements	Quantity (in mg/L)
Ca (NO_3_ 4H_2_O)	10.48	H_3_ BO_3_	1.14
NH_4_ H_2_ PO_4_	1.8	Mn Cl_2_·4H_2_O	0.72
Mg SO_4_·7H_2_O	8	Zn SO_4_·7H_2_O	0.088
KNO_3_	9	Cu SO_4_·5H_2_O	0.064
		H_2_MoO_4_·H_2_O	0.014
EDTA	5.22 g		
FeSO_4_,7H_2_O	4.98 g		

## Data Availability

Not applicable.

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
