# Peer review of "Effects of Irrigation with Microcystin-Containing Water on Growth, Physiology, and Antioxidant Defense in Strawberry Fragaria vulgaris under Hydroponic Culture"

_toxins, 2022, doi:10.3390/toxins14030198_

Round 1
Reviewer 1 Report
The MS deals with a very interesting subject, the effects of cyanotoxins on strawberry growth, biochemical and physiological characteristics in a simple hydroponic culture. A lot of work has been done since a variety of measurements have been performed in leaves and roots of the plants exposed to 4 MCs levels, including two harvests at 30 and 60 days after the treatment commencement.
The English language is in a good level, some minor corrections are noted below.
My most serious concerns are about the methodology and the presentation of the results which to my opinion weakens a lot the conclusions drawn. Specifically, the issues raised are the following:
Methodology
- It is not clear how the authors measured the concentration of MCs in their cyanobacterial material. As stated in L433-437 the performed a PP2A inhibition assay. There are some issues about that:
- This is not a common method to measure concentration of MCs. The authors should better describe it.
- The reference of the method seen in L435 (Bouaïcha et al. 2001) does not exist in the Literature.
- The authors should justify the connection between PP2A inhibition assay and MC-LR concentration.
- And why particularly MC-LR, how do they conclude that their crude extract contains this variant.
- There is a misuse of the term “photosynthesis” throughout the MS. Photosynthesis is not measured in this study. Stomatal conductance is related with photosynthetic process, but it is NOT photosynthesis. The same holds for chlorophyll fluorescence. It relates to the efficiency of photosystem II, but it is NOT photosynthesis. Please use the exact terms of what is measured and avoid misleading groupings of parameters.
- PCA (principal component analysis) is not justified, not explained, not discussed in the MS. I really cannot understand why the authors have chosen to include it and why they do not use its output. The text below the PCA title in the Results section (L267-273) is irrelevant and in fact is a copy-paste from L219-224. The PCA figure is not discussed anywhere and the only reference to PCA is in the last sentence of the conclusions.
Presentation of results
- The presentation of the results in the relevant section has dissimilarities with the actual numbers and differences shown in Figures and Table. This holds for almost every figure. As a result, it is frequently stated that the x parameter was reduced in a concentration-dependent manner, but this is not supported by the figure and the statistical differences depicted there. Please see my specific comments below.
- There are many problems with the statistical signs of Table and Figures, some of them are obviously wrong, please see my specific comments below.
The result of the above-mentioned serious methodological and presentation-related issues is that the Discussion and the Abstract should be re-written after the issues are solved. Authors must go through them thoroughly.
Other major corrections
L82-84: Not precise sentence for two reasons: 1) only growth parameters are presented in Table 1 and no "photosynthesis", b) concentration did affect RFW and LFW although exposure did not. The sentences of L81-88 should be re-written for a more correct presentation of the actual results.
L89-94. Not correct description of the results: In Harvest 1 RL, RDW, NL, LFW and in Harvest 2 RL, NL and LFW does not show statistically significant decreases in a concentration dependent manner!
Table 1:
1) in all figures the treatments are presented as C, C1, C2...
2) I do not find any good reason to statistically compare the treatment C2-Harvest 1 with C4-Harvest2, thus the post-hoc tests should be performed between the treatments of the same Harvest-for all the other comparisons two-way ANOVA is fine,
3) I am not sure that the statistical signs are correct: eg in RL-Harvest 1 what is the meaning of c sign in 10um/L??
L133-138: the statistical results presented for carotenoids in Fig 2d does not support the concentration-dependent reduction stated here.
L148-149: this is not correct, since a) in Harvest 1 the C1, C3 and C4 are the same and b) C1, C2 and C3 showed the same MDA conc in Harvest 2 which was certainly not significantly different as stated here.
L173-176: here should be clearly stated that there is not significant concentration-dependent increases. Of course the correction of the statistical signs in Fig 4 will help reader's understanding of the true differences.
Figure 1b: the statistical signs are wrong. Which treatment has b' ?
Figure 4a: the statistical signs are wrong. In Harvest 1which is (only) b here? C, C1 and C3 do not differ, but they all differ from C2 and C4. Thus, the correct signs are: a, a, b, a, b.
Figure 4b: the statistical signs are wrong. Which is (only) b' here?
Figure 5a: the statistical signs are wrong. Please correct.
L450-454. what leaf length mean in a three parts leaf as the strawberry leaf. Please explain better what was measured for leaf and root length. Additionally, the fresh weight of the root is a measurement of doubtful accuracy, since the thin root loses rapidly water and in case of prior washing of roots in order for the perlite to be removed, one cannot be sure whether the weight measured is solely the weight of the root or part of the washing water.
Minor corrections
L43: due to
L89-94: the abbreviations are explained the first time they appear and afterwards only the abbreviations should be used.
Table 1: In caption and explanation of parameters, please remove “stomatal conductivity” and “fluorescence” because they are NOT presented in this table.
F. vulgaris in italics throughout the text, e.g. L104, caption of Fig 1.
L132: what do you mean with “correlation”?
L166 and 181 please correct the word Table.
L84 and 87 correct leave to leaf or leaves.
Author Response
The MS deals with a very interesting subject, the effects of cyanotoxins on strawberry growth, biochemical and physiological characteristics in a simple hydroponic culture. A lot of work has been done since a variety of measurements have been performed in leaves and roots of the plants exposed to 4 MCs levels, including two harvests at 30 and 60 days after the treatment commencement.
The English language is in a good level, some minor corrections are noted below.
R: A deep revision of the manuscript text has been performed in order to correct all grammar mistakes.
My most serious concerns are about the methodology and the presentation of the results which to my opinion weakens a lot the conclusions drawn. Specifically, the issues raised are the Following :
Methodology
- It is not clear how the authors measured the concentration of MCs in their cyanobacterial material. As stated in L433-437 the performed a PP2A inhibition assay. There are some issues about that:
- This is not a common method to measure the concentration of MCs. The authors should better describe it.
R: More detail of the method description has been added to the text (Lines 486 to 494). According to reference Bouaïcha et al. (2001) [59]
- The reference of the method seen in L435 (Bouaïcha et al. 2001) does not exist in the Literature.
R: The citation of the reference (Bouaïcha et al. 2001) has been added in the literature list (Lines 797 to 798).
- The authors should justify the connection between PP2A inhibition assay and MC-LR concentration.
R: More information related to the relationship between the PP2A inhibition test and the concentration of MC-LR is justified (Lines 488 to 489).
- And why particularly MC-LR, how do they conclude that their crude extract contains this variant.
R: About the dominance of MC-LR variant (98%) in the Microcystis bloom extract was already analyzed and published in El Khalloufi et al. (2013 [58]. This information was added to the text (Line 491 to 492).
- There is a misuse of the term “photosynthesis” throughout the MS. Photosynthesis is not measured in this study. Stomatal conductance is related with photosynthetic process, but it is NOT photosynthesis. The same holds for chlorophyll fluorescence. It relates to the efficiency of photosystem II, but it is NOT photosynthesis. Please use the exact terms of what is measured and avoid misleading groupings of parameters.
R: YES completely agree with you professor!, the text has been changed as recommended (Line 141)
- PCA (principal component analysis) is not justified, not explained, not discussed in the MS. I really cannot understand why the authors have chosen to include it and why they do not use its output. The text below the PCA title in the Results section (L267-273) is irrelevant and in fact is a copy-paste from L219-224. The PCA figure is not discussed anywhere and the only reference to PCA is in the last sentence of the conclusions.
R: More information about the statistical tool PCA was added in order to justify the useful of the analyses . The modification has been made (Lines 302 to 316) also in data analysis in materiel and methods section.
Presentation of results
- The presentation of the results in the relevant section has dissimilarities with the actual numbers and differences shown in Figures and Table. This holds for almost every figure. As a result, it is frequently stated that the x parameter was reduced in a concentration-dependent manner, but this is not supported by the figure and the statistical differences depicted there. Please see my specific comments below.
There are many problems with the statistical signs of Table and Figures, some of them are obviously wrong, please see my specific comments below
R: The corrections were performed according to your suggestions. Many Thanks
The result of the above-mentioned serious methodological and presentation-related issues is that the Discussion and the Abstract should be re-written after the issues are solved. Authors must go through them thoroughly.
R: The modifications were performed and the great part of the MS were modified and revised (Lines 33 to 46 for abstract and 331 to 460).
Other major corrections
L82-84: Not precise sentence for two reasons: 1) only growth parameters are presented in Table 1 and no "photosynthesis", b) concentration did affect RFW and LFW although exposure did not. The sentences of L81-88 should be re-written for a more correct presentation of the actual results.
1) R: - The modifications were performed in (Table 1)
- The sentences were corrected as suggested for better understanding (Lines 100 to 111)
L89-94. Not correct description of the results: In Harvest 1 RL, RDW, NL, LFW and in Harvest 2 RL, NL and LFW does not show statistically significant decreases in a concentration dependent manner!
R: The paragraph was totally changed and modified according to the earlier comments. The modifications considered the results related to Crop 1: RL, RDW, NL, LFW and to Crop 2: RL, NL and LFW that showed significant differences (Table 1) (Lines 100 to 111).
Table 1:
in all figures the treatments are presented as C, C1, C2...
R: The indication of treatments has been changed from T, C, C1, C2, C3, and C4 to 0 μg/L, 1 μg/L, 5 μg/L, 10 μg/L, and 20 μg/L (Modifications made in all figures)
- I do not find any good reason to statistically compare the treatment C2-Harvest 1 with C4-Harvest2, thus the post-hoc tests should be performed between the treatments of the same Harvest. For all the other comparisons two-way ANOVA is fine,
R: The statistical analysis have been redone and verified and the Table 1 has been updated.
- I am not sure that the statistical signs are correct: eg in RL-Harvest 1 what is the meaning of c sign in 10um/L??
R: Yes! the statistical signs (letters) were checked and modified and the Table 1 has been updated.
L133-138: the statistical results presented for carotenoids in Fig 2d does not support the concentration-dependent reduction stated here.
R: The modification generated by the correction of the statistical analysis, it is obvious that the interoperation of the results at Fig 2 is modified. Thus, the description of the results have undergone a change. (Line 161 to 165)
L148-149: this is not correct, since a) in Harvest 1 the C1, C3 and C4 are the same and b) C1, C2 and C3 showed the same MDA conc in Harvest 2 which was certainly not significantly different as stated here.
R: The correction was operated and the explanation has been changed according the results (Line 176 to 180)
L173-176: here should be clearly stated that there is not significant concentration-dependent increases. Of course the correction of the statistical signs in Fig 4 will help reader's understanding of the true differences.
R: The correction has been made (Line 200 to 206). In addition, the statistical signs were corrected in figure 4 to make clear and understandable the recorded difference
Figure 1b: the statistical signs are wrong. Which treatment has b' ?
R: The statistical signs were corrected (Figure 1b)
Figure 4a: the statistical signs are wrong. In Harvest 1which is (only) b here? C, C1 and C3 do not differ, but they all differ from C2 and C4. Thus, the correct signs are: a, a, b, a, b.
R: The statistical analysis of the results was verified and the signs were corrected (Figure 4)
Figure 4b: the statistical signs are wrong. Which is (only) b' here?
R: The statistical analysis of the results was verified and the signs were corrected (Figure 4)
Figure 5a: the statistical signs are wrong. Please correct.
R: The statistical analysis of the results was verified and the signs were corrected (Figure 5)
L450-454. what leaf length mean in a three parts leaf as the strawberry leaf. Please explain better what was measured for leaf and root length. Additionally, the fresh weight of the root is a measurement of doubtful accuracy, since the thin root loses rapidly water and in case of prior washing of roots in order for the perlite to be removed, one cannot be sure whether the weight measured is solely the weight of the root or part of the washing water.
Three leaflets measuring from 10 to 20 cm high and constituting the aerial part form the strawberry plants. It also has a large rhizome (underground part), and the two parts are well visible and separated.
Concerning perlite washing, this substrate is very easy to remove and does not require a large amount of water. Plants are quickly washed and then the excess of water is removed in order to measure the lengths and weights.
Minor corrections
L43: due to
R: The proposed sentence has been added (Line 6).
L89-94: the abbreviations are explained the first time they appear and afterwards only the abbreviations should be used.
R: The remark was taken into consideration in the MS as proposed
Table 1: In caption and explanation of parameters, please remove “stomatal conductivity” and “fluorescence” because they are NOT presented in this table.
R: The proposed correction was made (Line 117 and caption of Fig 1)
- vulgarisin italics throughout the text, e.g. L104, caption of Fig 1.
R: The scientific name was corrected as recommended (Line 124 to 125)
L132: what do you mean with “correlation”?
R: The word was changed by interaction. The meaning is that the interaction between the same concentration at both harvests is highly significant (Line 38)
L166 and 181 please correct the word Table.
R: The suggested correction has been made (Line 167 and 192)
L84 and 87 correct leave to leaf or leaves.
R: The suggested correction has been made (Line 84 and 87)
Reviewer 2 Report
the paper needs some modifications in the results section line 93 all treatments caused significantly decreased not the high concentrations only
please clarify ug/l or ug L-1 in all manuscript
in the legend of figures please write what mean by c c1 c2 treatments
in chl a also c1 caused non-significant effect
the references are not in the format of the journal especially in the discussion section
Author Response
The paper needs some modifications in the results section line 93 all treatments caused significantly decreased not the high concentrations only
R: The statistical analysis has been corrected and the text was modified accordingly (Line 100 to 108).
please clarify ug/l or ug L-1 in all manuscript
R: The modification was conducted in the all manuscript
in the legend of figures please write what mean by c c1 c2 treatments
R: The remark was considered and figures were modified
in chl a also c1 caused non-significant effect
R: The modification was considered (Line 145 to 146).
the references are not in the format of the journal especially in the discussion section
R: The references citation was modified according to journal guidelines as recommended.
Reviewer 3 Report
The manuscript presented is about the impact of irrigation with MCs in the strawberry. The results presented are interesting. However, there are several details that must be improved to increase the quality of the manuscript.
General comments:
In my opinion, you should not perform a two-way ANOVA in the study. The recommendation is to perform one way ANOVA for each sampling moment. This will simplify the presentation of the results and it will be easier to understand them.
The discussion must be improved substantially. The authors mostly only make comparisons of their results with the results of the literature. However, they do not discuss the mechanisms by which the results are obtained.
Uniformize the use of supervisors and subscripts in the units. Check the entire manuscript.
All scientific names must be in italic.
"ml" should be replaced by "mL". Review throughout the document.
Specific comments:
Lines 73-74. The authors include information that is part of the methodology in the objective. In my opinion it is not necessary to mention the concentrations and the time of development of the crop.
Line 95. In the description of Table 1 the authors use asterisk to show significant diferences.
Line 101. SFW should be LFW?
Lines 266-273. This is the description of Figure 7, but not the PCA. I think it's a mistake. In addition, no description of the PCA is presented.
Line 477. rpm must be expressed in xg. Review throughout the document.
Line 548. It is not necessary to include the word "analysis" after ANOVA.
Author Response
The manuscript presented is about the impact of irrigation with MCs in the strawberry. The results presented are interesting. However, there are several details that must be improved to increase the quality of the manuscript.
General comments:
In my opinion, you should not perform a two-way ANOVA in the study. The recommendation is to perform one way ANOVA for each sampling moment. This will simplify the presentation of the results and it will be easier to understand them.
R: The objectives of the conducted work was the evaluation of the effect of Microcystins concentrations (0, 1, 5, 10, 20 µg/L) on F. vulgaris parameters and then the effect of the exposure time (30 days and 60 days). So we based the statistical analysis on two factors (concentration and exposure time).
The discussion must be improved substantially. The authors mostly only make comparisons of their results with the results of the literature. However, they do not discuss the mechanisms by which the results are obtained.
R: The discussion part was improved as suggested (all discussion).
Uniformize the use of supervisors and subscripts in the units. Check the entire manuscript.
R: The modification was conducted in the all manuscript
All scientific names must be in italic.
R: The modification was conducted in the all manuscript
"ml" should be replaced by "mL". Review throughout the document.
R: The modification was considered throughout the manuscript
Specific comments:
Lines 73-74. The authors include information that is part of the methodology in the objective. In my opinion it is not necessary to mention the concentrations and the time of development of the crop.
R: The information has been deleted (Line 92 to 95).
Line 95. In the description of Table 1 the authors use asterisk to show significant diferences.
R: Significant differences between microcystins concentrations effects were showed using letters and significant differences due to exposure time were indicated with asterisk. Both indications were considered since we have made two different analyses.
Line 101. SFW should be LFW?
R: The modification was conducted (Line 120)
Lines 266-273. This is the description of Figure 7, but not the PCA. I think it's a mistake. In addition, no description of the PCA is presented.
R: The modification was conducted as recommended (Lines 303 to 116)
Line 477. rpm must be expressed in xg. Review throughout the document.
R: The proposed correction was considered (all manuscript)
Line 548. It is not necessary to include the word "analysis" after ANOVA.
R: The proposed correction was considered (all manuscript)
Reviewer 4 Report
Introduction:
The introduction section is relatively good; however, there are some spaces for authors to enhance its quality further, which are as follows:
Improve the introduction by inserting more information about using microcystins with a nutrient solution in hydroponics and why using the substrate culture(perlite) is not deep-water culture (DWC) or nutrient film technique(NFT).
In lines 71 until 74: The objective of the present study was to examine the effects of long-term irrigation with an extract of the toxic Microcystis aeruginosa-bloom on the growth, biochemical parameters, and antioxidant response of strawberry plants exposed to four concentrations of microcystins (1, 5, 10, 20 μg/L) for 60 days in hydroponic culture it is an important objective, but there are other data must be collected to show the exact effect of toxic Microcysti on strawberry plants such as the fruit yield, quality and leaf nutrients contents.
Materials and Methods
Subtitles: Microcystins extraction and quantification
Please insert a picture for Microcystins extraction in the final phase before using with nutrient solution.
Subtitle: Experimental setup
In lines 444 until 447:
It is a bit difficult to understand your nutrient solution setup in this experiment, MCs extract was diluted with distilled water or Hoagland nutrient solution [62] to final concentrations: C= 0 μg/L, C1= 1 μg/L, C2= 5 μg/L, C3= 10 μg/L, and C4= 20 μg/L. Plants were watered with nutrient solution and water containing MCs at 2-days intervals, alternatively.
What about the electrical conductivity (EC), pH measurement?
Please insert a table that represents the standard chemical composition of the synthetic nutrient (Hoagland solution) applied to the strawberry plants Subtitle:
Please insert a schematic diagram of the nutrient solution tank supplying to the treatments of strawberry plants.
Results
Effects of microcystins on strawberry growth parameters
Please insert the figure for the strawberry experiment in the greenhouse under controlled environmental conditions after 30 and 60 days.
Table 1 shows the effect of microcystins on plants growth parameters, stomatal conductivity and fluorescence, it was shown that the LFW in the 1st harvest at 20 µg. L-1 MC was 12.63 g which is higher than 10 µg. L-1 at same harvest although the Number of leaves and Leaves length was lower at 20 µg. L-1 MC?
Can the authors explain the importance of parameter leaf length for strawberries in this study?
I hope the authors insert a correlation between the treatments (concentration of MC) and leaf number, root fresh weight etc.
Discussion
The discussion section must be presented under certain subtitles as the authors did for results. Meaning as authors presented their results under certain subtitles in Results, so are they suggested developing subtitles under the Discussion section too
References
The scientific name of the plant must be italic or put lines under the scientific name such as reference number [22] [26][27][29][30][34][35][38][40][45][46][66]
Author Response
Introduction:
The introduction section is relatively good; however, there are some spaces for authors to enhance its quality further, which are as follows:
Improve the introduction by inserting more information about using microcystins with a nutrient solution in hydroponics and why using the substrate culture (perlite) is not deep-water culture (DWC) or nutrient film technique (NFT).
R: The modifications were added as recommended (Lines 66 to 68)
In lines 71 until 74: The objective of the present study was to examine the effects of long-term irrigation with an extract of the toxic Microcystis aeruginosa-bloom on the growth, biochemical parameters, and antioxidant response of strawberry plants exposed to four concentrations of microcystins (1, 5, 10, 20 μg/L) for 60 days in hydroponic culture it is an important objective, but there are other data must be collected to show the exact effect of toxic Microcysti on strawberry plants such as the fruit yield, quality and leaf nutrients contents.
R: We totally agree with you, the data concerning the fruit quality control are part of our future work.
Materials and Methods
Subtitles: Microcystins extraction and quantification
Please insert a picture for Microcystins extraction in the final phase before using with nutrient solution.
R: Figure describing the steps of preparing the Microcystins extracts has been added (Line 476 to 497)
Subtitle: Experimental setup
In lines 444 until 447:
It is a bit difficult to understand your nutrient solution setup in this experiment, MCs extract was diluted with distilled water or Hoagland nutrient solution [62] to final concentrations: C= 0 μg/L, C1= 1 μg/L, C2= 5 μg/L, C3= 10 μg/L, and C4= 20 μg/L. Plants were watered with nutrient solution and water containing MCs at 2-days intervals, alternatively.
R: The nutrient solution was prepared from three stock solution: macroelements; microelements; EDTA-Fe. The irrigation of the plants was done 3 times/week alternating between distilled water (2 times) and the nutritive solution (water + essential elements) once.
What about the electrical conductivity (EC), pH measurement?
R: The optimal pH of the nutrient solution for normal development of strawberry plants should be between 5.7 and 6.5, our nutrient solution had a pH of 6.3. Concerning the electrical conductivity of our nutrient solution was between 1.66 and 2 mS/cm.
Please insert a table that represents the standard chemical composition of the synthetic nutrient (Hoagland solution) applied to the strawberry plants Subtitle:
R: The recommended table have been added (Line 514 to 515)
Please insert a schematic diagram of the nutrient solution tank supplying to the treatments of strawberry plants.
R: During the experiment period, strawberry plants were supplied with nutrients solution and Microcystins concentrations manually.
Results
Effects of microcystins on strawberry growth parameters
Please insert the figure for the strawberry experiment in the greenhouse under controlled environmental conditions after 30 and 60 days.
R: Figure of strawberry experiment under controlled condition was added as proposed (Line 520 to 532)
Table 1 shows the effect of microcystins on plants growth parameters, stomatal conductivity and fluorescence, it was shown that the LFW in the 1st harvest at 20 µg. L-1 MC was 12.63 g which is higher than 10 µg. L-1 at same harvest although the Number of leaves and Leaves length was lower at 20 µg. L-1 MC?
R: According to the statistical analysis the LFW and LN in the 1st harvest at 10 and 20 µg. L-1 MC showed no significant difference. However, for the LL parameters the decrease was significant at 20 µg. L-1 MC.
Can the authors explain the importance of parameter leaf length for strawberries in this study?
Yes, with pleasure, strawberry is characterized by the absence of stem, but the leaves (petiole and three leaflets) have a height of 10 to 20 cm, constituting all the aerial part; the effect of microcystins is well visible visually on this length of this part, so the measurement of the length of the leaves remains an easy tool to determine the effect of the MC on growth
I hope the authors insert a correlation between the treatments (concentration of MC) and leaf number, root fresh weight etc.
R: The proposed recommendation has been added (Line 318 to 319)
Discussion
The discussion section must be presented under certain subtitles as the authors did for results. Meaning as authors presented their results under certain subtitles in Results, so are they suggested developing subtitles under the Discussion section too.
R: The proposed modification was considered (all discussion)
References
The scientific name of the plant must be italic or put lines under the scientific name such as reference number [22] [26][27][29][30][34][35][38][40][45][46][66]
R: The correction was conducted throughout the manuscript
Round 2
Reviewer 1 Report
The revised MS is improved and most of my remarks were addressed.
Nevertheless, there are serious issues that were overlooked in R1:
- Although the authors agreed with my comment that they do NOT measure photosynthesis, but related to it processes, they have not removed the word photosynthesis neither from the TITLE of the MS nor from Abstract, M+M and Conclusions. Please do change the title and replace the word throughout the MS.
- The authors do not explain or justify the connection between PP2A inhibition assay and MC-LR concentration. In their response to my comments, they claim that they have included the related information, but the only relevant text I could read was: “The PP2A analysis showed a total MCs concentration of about 11.5 mg MC-LR/g DW”. This is not an explanation. Additionally, the lines they refer to (Lines 488 to 489) contain information on growth chamber conditions (in the pdf I received) and not the text they claim to have added.
Minor corrections
- L30-31: 10 and 20 μg/L cannot be considered “low concentration” but rather high compared to the published environmentally relevant dosages. Additionally, no comparison should be made with soil culture, especially in the Abstract since nothing like that was determined or examined in the study. Please modify the whole sentence.
- Figure 1: no reference in the caption to two-way ANOVA
- Please remove the L. after the scientific name of the plant throughout the MS. You should keep it only the first time you give the full name in the Abstract (L11). Afterwards, only F. vulgaris should be written.
Author Response
Reviewer 1:
The revised MS is improved and most of my remarks were addressed.
Nevertheless, there are serious issues that were overlooked in R1:
- Although the authors agreed with my comment that they do NOT measure photosynthesis, but related to it processes, they have not removed the word photosynthesis neither from the TITLE of the MS nor from Abstract, M+M and Conclusions. Please do change the title and replace the word throughout the MS.
R: The modification was conducted in the all manuscript:
- From the title (Line 3).
- Abstract (Line 29 and 46).
- Results (Line 145).
- Principal Component Analysis section (Line 304, 308, 310, 312 and 321)
- Discussion (383 and 384).
- Materials and Methods (Line 531 and 540).
- The authors do not explain or justify the connection between PP2A inhibition assay and MC-LR concentration. In their response to my comments, they claim that they have included the related information, but the only relevant text I could read was: “The PP2A analysis showed a total MCs concentration of about 11.5 mg MC-LR/g DW”. This is not an explanation. Additionally, the lines they refer to (Lines 488 to 489) contain information on growth chamber conditions (in the pdf I received) and not the text they claim to have added.
A: In order to meet the reviewer's expectations. Some technical details have been added to the paragraph on "extraction and quantification of microcystins". This part has been completely rewritten. The section describing the PP2A performed has also been improved to clarify the principle of the test performed. The MC contained in the bloom extract inhibits PP2A and consequently the dephosphorylation of para-nitrophenylphosphate (p-NPP) is also affected which induces a reduction of the colored product: the formation of para-nitrophenol (Line 479 to Line 498)
Minor corrections
- L30-31: 10 and 20 μg/L cannot be considered “low concentration” but rather high compared to the published environmentally relevant dosages. Additionally, no comparison should be made with soil culture, especially in the Abstract since nothing like that was determined or examined in the study. Please modify the whole sentence.
- R: The correction was considered in the all manuscript.
- Figure 1: no reference in the caption to two-way ANOVA.
- R: The modification has been made.
- Please remove the L. after the scientific name of the plant throughout the MS. You should keep it only the first time you give the full name in the Abstract (L11). Afterwards, only F. vulgaris should be written.
- R: the suggested correction has been made in the all manuscript.
Reviewer 3 Report
The authors have made various changes to the manuscript that improved some aspects. However, as far as statistical methods are concerned, there are still shortcomings.
It is unclear how the ANOVAs were performed. This in turn affects the presentation of the results, and makes interpretation difficult.
In the Figures it is not clear how the comparison between the evaluated treatments and the exposure time is made. Authors use letters and symbols.
Author Response
The authors have made various changes to the manuscript that improved some aspects. However, as far as statistical methods are concerned, there are still shortcomings.
It is unclear how the ANOVAs were performed. This in turn affects the presentation of the results, and makes interpretation difficult.
In the Figures it is not clear how the comparison between the evaluated treatments and the exposure time is made. Authors use letters and symbols.
We are grateful for your useful and beneficial comments and suggestions that we took into consideration in order to improve the clarity of the manuscript.
R: We added a text description in the data analysis section 5.2, to clearly indicate how we performed the Two-way ANOVA using SPSS (Line 622 to 632).
In addition, we changed the titles of the figures to make them easier to read in the all manuscript.
Round 3
Reviewer 3 Report
The manuscript has improved with the changes made. In my opinion, the manuscript can be accepted for publication.